# Object-Specific Multiview Classification Through View-Compatible Feature Fusion

**DOI:** 10.3390/s25134127

**Published:** 2025-07-02

**Authors:** Javier Perez Soler, Jose-Luis Guardiola, Nicolás García Sastre, Pau Garrigues Carbó, Miguel Sanchis Hernández, Juan-Carlos Perez-Cortes

**Affiliations:** 1Instituto Tecnológico de Informática (ITI), C. Nicolás Copérnico, 7, 46022 Valencia, Spain; 2Departamento de Informática de Sistemas y Computadores (DISCA), Universitat Politècnica de València (UPV), 46022 Valencia, Spain

**Keywords:** multi-view classification, feature fusion, industrial inspection, open-set classification

## Abstract

Multi-view classification (MVC) typically focuses on categorizing objects into distinct classes by employing multiple perspectives of the same objects. However, in numerous real-world applications, such as industrial inspection and quality control, there is an increasing need to distinguish particular objects from a pool of similar ones while simultaneously disregarding unknown objects. In these scenarios, relying on a single image may not provide sufficient information to effectively identify the scrutinized object, as different perspectives may reveal distinct characteristics that are essential for accurate classification. Most existing approaches operate within closed-set environments and are focused on generalization, which makes them less effective in distinguishing individual objects from others. This limitations are particularly problematic in industrial quality assessment, where distinguishing between specific objects and discarding unknowns is crucial. To address this challenge, we introduce a View-Compatible Feature Fusion (VCFF) method that utilizes images from predetermined positions as an accurate solution for multi-view classification of specific objects. Unlike other approaches, VCFF explicitly integrates pose information during the fusion process. It does not merely use pose as auxiliary data but employs it to align and selectively fuse features from different views. This mathematically explicit fusion of rotations, based on relative poses, allows VCFF to effectively combine multi-view information, enhancing classification accuracy. Through experimental evaluations, we demonstrate that the proposed VCFF method outperforms state-of-the-art MVC algorithms, especially in open-set scenarios, where the set of possible objects is not fully known in advance. Remarkably, VCFF achieves an average precision of 1.0 using only 8 cameras, whereas existing methods require 20 cameras to reach a maximum of 0.95. In terms of AUC-ROC under the constraint of fewer than 3σ false positives—a critical metric in industrial inspection—current state-of-the-art methods achieve up to 0.72, while VCFF attains a perfect score of 1.0 with just eight cameras. Furthermore, our approach delivers highly accurate rotation estimation, maintaining an error margin slightly above 2° when sampling at 4° intervals.

## 1. Introduction

Multiview classification (MVC) and pose estimation have become essential components in computer vision systems, enabling robust understanding of scenes and human activities from multiple viewpoints. Rather than relying on a single perspective, MVC combines data captured from different angles to enhance classification accuracy and resilience to complex and similar geometries, occlusions, lighting variation, and other visual ambiguities. Pose estimation complements this by providing detailed spatial information about object or body configurations, which is particularly useful when integrated across multiple views.

These technologies are increasingly critical across a wide range of applications. In sports analytics, multiview systems facilitate precise tracking of athletes and equipment, supporting real-time strategy refinement and performance assessment [1,2]. In surveillance, MVC enables reliable person identification and activity recognition in complex environments where single-view systems often fail [3]. The entertainment industry employs MVC and pose estimation in animation and virtual reality for creating immersive and lifelike digital experiences [4]. In medicine, multiview data can enhance remote monitoring for chronic conditions through rehabilitation exercise [5]. In the work of Xuan et al. [6] Generative Adversarial Networks were used to expand the training set in a multiview setup to classify pearls. Furthermore, human–computer interaction systems exploit MVC and pose estimation to build intuitive interfaces that respond to gestures and body language [7].

In the industrial inspection systems, where this study is aimed, MVC and pose estimation play a pivotal role, given that a single system is tasked with analyzing a wide array of references, including mixtures [8]. Moreover, manufactured objects often manifest similar variations, such as small changes or mirrored references, posing challenges for identification on single cameras or even by expert automated inspection frameworks [9]. They have become indispensable tools for enhancing defect detection and quality assurance processes [10]. By capturing and analyzing data from multiple viewpoints, these technologies address challenges such as occlusions, varying lighting conditions, and complex geometries that often hinder single-view inspection systems. Moreover, within this context, the problem is inherently open-set, meaning that not all object classes are available during training. As a result, the system must not only accurately classify known objects but also effectively identify and reject previously unseen or unknown instances, thereby increasing the complexity of the task. Figure 1 illustrates a case where multiple views of an object are utilized.

This study introduces a novel approach that centers on a key innovation: the explicit integration of pose information into the classification process. Rather than treating classification and pose estimation as separate tasks, the method unifies them, leveraging the inherent rotation and position information computed during classification. This integration enables more precise fusion of multiview data by selectively combining compatible poses, outperforming traditional pose-agnostic methods. Experimental results confirm that incorporating pose information significantly enhances classification accuracy, especially in complex scenarios, and comparisons with two state-of-the-art methods highlight the advantages of explicitly using pose data.

The main contributions of this work are:A mathematically explicit method for fusing evidence from multiple cameras: View-Compatible Feature Fusion (VCFF). This approach takes into consideration only feasible combinations of positions; thus, actually taking into consideration the third dimension instead of a combination of 2D assesments.Seamless integration of pose into the multi-view classification process, enabling the extraction of precise conclusions from multi-view observations.The classification process yields pose estimation as a side result, in our experiments this pose estimation reached 2° average error which is a great result as we are sampling at 4° intervals.The proposed method is capable of easily detecting objects not present during the training as unknowns, critical feature in industrial inspection, achieving perfect 1.0 average precision even in an open-set configuration.Addition and removal of objects in the training set is fast and efficient. In the case of adding new classes to train, it is just necessary to train the introduced class instead of performing a complete retraining.New multi-view classification dataset focused on the industrial inspection context including similar objects, mirrored versions, unknown references, as well as rotation ground truth for classification and pose estimation.

The paper is organized as follows: Section 2 offers a thorough review of the current state of Multi-View Classification (MVC). Section 3 elaborates on the datasets and algorithms proposed for feature extraction and fusion within this study. Section 4 presents the findings, comparing various state-of-the-art approaches. Finally, Section 5 summarizes the contributions made by this work and delineates potential future research lines.

## 2. State of the Art

In the pre-deep learning era, multi-view classification techniques were largely driven by handcrafted features, geometric descriptors, and statistical learning methods. Mainly, two different strategies prevailed, 3D shape descriptors and view-based approaches.

3D shape descriptors were typically represented as histograms of features such as distances, angles, triangle areas or tetrahedra volumes gathered at sampled surface points as explained in Osada et al. [11], properties of spherical functions defined in volumetric grids as in Kazhdan et al. [12], heat kernel signatures on polygon meshes as described in Shape [13] or Kokkines et al. [14] or even SIFT- and SURF-based such as in Knopp et al. [15]. Building a classification system over these descriptors presents different challenges, such as the limited amount of 3D databases, and the dimensionality curse due to the high dimensionality of these descriptors.

On the other hand, view-based descriptors are relatively low dimensional, efficient to evaluate and have a large amount of images to train. One of the first examples of multi-view classification is described in Murase et al. [16], it learns from multiple renders under varying poses and illuminations. Chen et al. [17] rely on geometric and Fourier descriptors. Eitz et al. [18] used Gabor filters to compare human sketches with line drawings of 3D models. Unfotunately all these descriptors were ignored when neural networks provided meaningful descriptors capable of generalizing well across different domains and obtaining a significantly higher performance.

Since the emergence of deep learning and the seminal work by Su et al. on Multi-View Convolutional Neural Networks (MVCNN) [19], the recognition of 3D shapes has predominantly favoured multiview approaches over 3D methods. A comprehensive review by Mona et al. [20] and especially Guo et al. [21] highlight this trend, encompassing voxel-based methods such as those proposed by Wang et al. [22], and point-based methodologies as exemplified by Zhang et al. [23] and Fernandes et al. [24].

These multiview approaches leverage diverse and complementary information from different angles of the same data, enabling efficient problem-solving. In contrast, 3D information often presents challenges due to its sparsity and the complexities involved in representation and processing. Furthermore, real-world applications frequently involve input from multiple cameras, as demonstrated in industrial inspection scenarios as discussed in Ren et al. [25] or such as the inspection device described by Perez-Cortes et al. [26], while 3D information is typically scarce.

Multi-view-based techniques, extensively reviewed in multiple reviews [20,27,28,29], typically exhibit three key characteristics: viewpoint input, feature extraction, and feature fusion.

The viewpoint input stage defines how views are incorporated into the algorithm and can be categorized into two primary groups: simultaneous or sequential. Simultaneous approaches, such as those proposed by Su et al. [19] and Wei et al. [30], allow the introduction of any number of views from arbitrary positions without constraints. However, they lack the capability to learn 3D information.

In contrast, sequential approaches rely on a series of views to infer the 3D structure from multiple perspectives. Examples of this approach include the works of Han et al. [31], Kanezaki et al. [32], and Chen et al. [33]. However, these methods require the same sequence of views during testing, which may not always be available. Some recent approaches attempt to overcome these limitations by incorporating attention mechanisms. For instance, Sun et al. [34] introduced a transformer-based method that allows processing views in any order. Despite this improvement, the method remains tied to a specific camera configuration. Alzahrani et al. [35] introduced cosine distance to select the most relevant view and filter information before classifying. The main drawback of this method is that it assumes only one view is enough to identify the object, which may not be true for highly similar objects.

Both approaches offer distinct advantages and drawbacks. In this study, we propose a hybrid model that leverages the strengths of both. Our approach utilizes a sequence of views during training while incorporating pose information, that can be generated through simulation. During testing, it relies on precomputed extrinsic camera positions. This strategy enables the use of 3D information without being restricted to a predefined set of views. The only limitation is the need to know the relative camera positions. However, this requirement is easily met in stable environments, such as industrial settings where cameras remain fixed.

The subsequent stage, feature extraction, delineates the image processing methodology. Typically, a well-performing 2D classification-task backbone network serves as the prevailing approach in this stage. For example, MVCNN [19] employed VGG-M [36], GVCNN [37] utilized GoogLeNet [38], and RotationNet [32] implemented AlexNet [39]. Other approaches such as Zhang et al. [40] crafted specific methods for highly similar sheet metal parts in grey pixels. However, in this study, we employ a Fourier Descriptor based on centroid distance during feature extraction, similar to Zhang et al. [41], as detailed in Section 3.3. This choice is motivated by a desire to maintain low complexity, ensuring a primary focus on the core innovation: view-compatible feature fusion. Notably, any other feature extractor could be used.

Moreover, the adoption of this straightforward feature extractor facilitates the establishment of unknown thresholds and the seamless addition of new objects to the training set without completely retraining the model, a valuable characteristic in industrial inspection applications. Additionally, the efficacy of this feature extractor in 2D shape representation and retrieval has been demonstrated in previous works, as evidenced by Zhang et al. [41]. The main drawbacks are the limited generalization, which may be desirable in the industrial inspection context to detect only specific objects, and limited complex feature extraction capabilities ignoring texture or color characteristics.

Lastly, feature fusion represents the concluding element of multi-view-based techniques, determining how information from all images is integrated. This stage is completely dependant on the viewpoint input. Initial endeavors, mainly used in simultaneous input, employed straightforward pooling techniques, as demonstrated in MVCNN [19], albeit with limited 3D discriminatory power. GVCNN [37] introduced group-level descriptions to incorporate information from various views, yet still relied on pooling mechanisms.

Recurrent networks, exemplified by Han et al. [31] and Jiang et al. [42], aim to address this challenge through sequences of images, with the aspiration of learning features in a 3D-wise manner. Alternatively, other approaches attempt to pool view-wise features using diverse mechanisms, such as the utilization of a harmonized pooling layer in Yu et al. [43], or employing multi-hypergraph learning techniques as demonstrated by Zhang et al. [44]. Rather than solely concentrating on enhancing pooling mechanisms, alternative approaches in the literature have shifted focus towards view selection, as demonstrated by Wei et al. [30]. RotationNet [32] introduced the concept of incorrect views to represent samples affected by occlusions, thereby addressing this issue. Current trends are focused on attention approaches such as ViewFormer [34].

While these fusion strategies typically yield good results across various scenarios, they often overlook readily available information in real-world settings: the estimation of relative camera locations. Such data can be acquired through multicamera calibration [45] in stationary camera setups, or via acceleration and magnetic sensors on mobile cameras [46]. Notably, RotationNet [32] tries to incorporate this information during training, offering a rough rotation estimation as a by-product of the classification process.

In this study, we propose a mathematically explicit solution for fusing camera rotations based on their relative poses to merge view-compatible features. We refer to this method as View-Compatible Feature Fusion (VCFF). As highlighted in reviews of multi-view classification techniques [27,28,29], this area offers significant opportunities for improvement. By employing this explicit fusion method, we achieve higher precision in distinguishing visually similar objects, such as mirrored versions, and improve the detection of unknowns.

## 3. Material and Methods

The proposed multi-view classification approach finds its primary application in industrial inspection, a domain characterized by three distinctive characteristics compared to generic 3D object classification. First, it does not require shape generalization, as the objects under inspection closely match their 3D models. Second, the reference objects or classes often exhibit subtle differences, making them inherently difficult to distinguish. Third, any object deviating from the 3D shape of known classes must be identified as unknown, framing the problem as an open-set classification challenge.

Although originally designed for industrial inspection applications, this method can be applied in other contexts—provided the necessary inputs are available, such as a 3D model and the relative positions of the cameras. However, it lacks generalization capability, as it focuses on identifying specific objects and the 3D models must closely resemble the real objects, making this method suitable only for use cases that demand high classification specifity.

Furthermore, in industrial inspection settings, unexpected object occlusions typically occur only when something goes wrong. As a result, they are not explicitly handled. Any occluded object is therefore treated as an unknown object. Although the proposed method may be capable of handling occlusions, this scenario is considered outside the scope of the current work.

The VCFF method introduced in this study operates under the premise that classifying a 3D object in an image inherently involves estimating its pose. Acknowledging this fundamental aspect, pose information is integrated into the training process by incorporating object rotation relative to the camera. This information can easily be added using realistic simulation as precise 3D models are usually available in the industrial inspection context. During testing, this information is encapsulated through the relative positions of cameras, allowing the translation and rotation of each camera result to any other camera point of view in search of coherence.

Figure 2 illustrates a complete view of VCFF and Figure 3 provides a closer look at VCFF. An example of view-compatible feature fusion using six cameras positioned at 45° intervals is shown. As depicted, each view extracts its own rotation-aware features, compares them with a model, and then aligns them within a unified feature space before merging them into a single representation. From this representation, it is possible to obtain the probability pertaining to the model as well as the most probable rotation of the object. The process is described in detail in the following Section 3.2, Section 3.3, Section 3.4 and Section 3.5 and pseudocode for classification is provided in Algorithm A2 in Appendix A.

### 3.1. Datasets

The unique requirements of industrial inspection applications reveal significant limitations in widely used public datasets such as ModelNet, ShapeNet, MIRO, and RGB-D. These datasets often necessitate substantial generalization to achieve classification, whereas industrial inspection demands the precise identification of specific reference objects from a pool of similar ones. Additionally, these datasets typically lack both: similar unknown objects and the relative camera position data required for the proposed approach.

For these reasons, a new dataset has been created, taking industrial parts as an example. This dataset includes objects with mirrored versions or subtle modifications of the industrial part as well as simple testing objects. Furthermore, similar objects have been designated as unknowns to rigorously evaluate the discrimination capabilities of the proposed approach. The dataset is publicly available at Zenodo (https://zenodo.org/records/15058296, accessed on 29 June 2025) [47]. It includes the 3D model of each object in the training set, as well as a test set consisting of 100 captures from 20 viewpoints per object, including unknown objects not present in the training set.

More precisely, the created dataset simulates punching scrap from plastic injection or thermoforming processes—such as the waste generated during the production of car dashboards. While these plastic pieces often share similar shapes across different car models, their material compositions can vary significantly. As a result, identifying the type of plastic is crucial for effective recycling, which in turn helps reduce both waste and production costs.

In this industry, only a limited variety of parts are typically produced at any given time, as creating and switching molds is an expensive and time-consuming process. However, many parts have mirrored or symmetrical counterparts—such as left and right seat components or steering wheel covers—and often feature similar design elements, like front and rear air conditioning vents.

The dataset summarized in Table 1 comprises 15 objects, 4 of which are unknown and thus not available for training. The objects include five with mirrored versions, one of these five is significantly smaller. The remaining five objects include two visually similar larger objects, a distinct object of regular size and two basic geometric shapes: a sphere and a cylinder. These objects were chosen to ensure the following use cases are correctly addressed:Distinguishing **mirrored objects**: Manufactured objects often have mirrored counterparts that must be identified, which can be challenging using a single camera.Identifying **similar objects outside the training** set: In industrial contexts, it is crucial to detect unknown objects, even if they closely resemble known ones. For this reason, a mirrored counterpart and an object of regular size have been kept out of the training set as unknown.Differentiating **outliers in the training set** from similar objects: Training datasets often include outliers that are easily classified. Due to inherent biases, some classifiers may fail to distinguish these outliers from similar objects outside the training set, as they do not prioritize learning the specific characteristics of such objects. To test this scenario, one of the larger objects was excluded from the training set and marked as unknown.Recognizing **unknown objects whose views overlap** with those of known objects: All views of a sphere, which appear as circles, overlap with the base view of a cylinder. However, despite this overlap, a collection of circle views from multiple points cannot fully reconstruct a cylinder. To evaluate this challenge, the sphere was excluded from the training set and labeled as unknown.

Figure 4 illustrates the objects included in the dataset. As can be seen, all the objects resemble industrial components. Objects designated as “unknown”, that are excluded from the training set are outlined in red. For training purposes, only the 3D model of each object is provided, allowing the proposed algorithms to utilize as many **simulated** images or projections as necessary.

### 3.2. Viewpoint Input

This stage aims to acquire multiple views alongside their respective pose information. However the imposition of fixed views at test could limit applicability in real-world scenarios. To address this, our approach involves employing fixed views during the training stage, utilizing simulated renders of the 3D object. During testing, any view can be utilized, but knowledge of the relative positions between views—referred to as extrinsic camera calibration—is crucial for accurate transformation. As discussed before, this information can be readily obtained. In scenarios where cameras remain stationary, such as in industrial inspection setups, multicamera calibration techniques can be used for this purpose. In situations involving moving cameras camera, relative poses can be inferred using motion sensors.

During the training stage, a set of image features or descriptors is extracted for each training view, resulting in a 3D matrix of feature descriptors. Each dimension accounts for each degree of freedom in rotation. This is the main innovation in this work: storing the pose information associated with each descriptor—captured from every sampled pose-allowing descriptors to be explicitly linked to their corresponding poses. The object is rotated with respect to the camera to generate image renders from various positions, facilitating feature extraction. See Algorithm A1 in Appendix A for more details.

In the case of rotation-invariant features, the camera rotation is redundant. Consequently, the dimensionality can be reduced to 2D to alleviate computational complexity without loss of generality or precision. This resulting 3D matrix of descriptors constitutes the trained model, which will be compared with test images during the feature fusion stage, so the most similar one can be identified.

Any rotation representation can be used, such as Euler angles, quaternions, and others. The choice of format affects properties like ease of rotation, dimensionality reduction, and interpretability. In this work, Euler angles were selected due to their intuitive interpretation. Specifically, the training features are represented as 2D matrices, with values ranging from 0 to 360° along the first axis and 0 to 180° along the second. Since the implementation assumes camera-rotation invariance, this dimension can be omitted during sampling to reduce computational complexity.

### 3.3. Feature Extraction and Comparison

Fourier Descriptors based on centroid distance were chosen for their simplicity and proven efficacy as a 2D shape descriptor, as demonstrated by Zhang et al. [41]. However, it is worth noting that any shape descriptor, including deep learning-based ones, could be utilized. These descriptors have notable limitations, such as relying only external shapes or ignoring textures or holes, which are discussed in Section 5. Despite this, they offer a key advantage: they allow effective training using simulated renders with only external outlines. This helps reduce the domain gap and avoids introducing additional complexity, keeping the focus of this work on the core innovation introduced by VCFF.

The process of extracting features from an image involves the following steps:Contour extraction: Equally spaced points are extracted from the contour of the object, forming a circular list that describes its 2D shape. This list of points can be smoothed to mitigate noise.Centroid distance calculation: The centroid of the object in the image is computed, and for each point in the contour list, the distance to this centroid is calculated, resulting in a 1D circular list of positive distances from the centroid.Scale invariance and filtering: The centroid distance function is transformed into the frequency domain and back using a Fast Fourier Transform (FFT). To filter out high-frequency noise, the number of components in the Fourier transform is limited. Additionally, the transformed function is divided by its first component to ensure scale invariance, allowing it to identify objects independently of their size in the image. Finally, it is transformed back into a fixed number of components for ease of comparison with other descriptors.

These operations effectively transform the image into a one-dimensional circular array of distances from the centroid, allowing for shape comparison to determine similarity with the test shape. This representation is inherently scale-invariant due to the use of frequency-domain operations. However, it lacks rotation invariance around the camera axis (i.e., image rotation invariance), which is desirable for reducing memory usage and computational complexity. As a result, directly comparing circular arrays is dependent on the object’s rotation within the image.

Some researchers achieve rotation invariance by disregarding phase information in the frequency domain, but this approach has been rejected, as it may remove crucial shape details. Instead, in this method, rotation invariance around the camera axis is achieved by shifting circular arrays to find the optimal alignment, minimizing the distance between them regardless of the object’s orientation. This phase adjustment can be efficiently performed in the frequency domain.

Figure 5 illustrates the features extracted for various shapes. In this depiction, the features for different shapes manifest distinct characteristics: a circle yields a straight line, a five-pointed star produces a sharp-edged signal with five regular cycles, and a rectangle generates four irregular soft cycles. These signals exhibit clear differentiation from one another.

The feature extraction process is applied to each image in the training set, which consists of a matrix of image renders across all possible rotations of the 3D model, as described in Section 3.2. Since the proposed image descriptors are rotation-invariant, the resulting matrix can be simplified by sampling only the remaining two dimensions, effectively reducing it to 2D. The trained model therefore consists of a 2D matrix of descriptors representing the reference object across all relevant rotations.

Each test image is then compared to this model using a distance or dissimilarity function, which involves matching the features extracted from the test image against those in the model.

A direct measurement of dissimilarity between aligned descriptors, such as standard L2 or L1 distances, can serve as an effective means of quantifying the disparity among shapes depicted in images. However, even a slight misalignment between descriptors, such as produced by minor variations in the 3D object or segmentation errors, as depicted in Figure 6, can result in significant distance discrepancies. To address this issue, we propose the use of the Dynamic Time Warp (DTW) metric, first introduced by Bellman in [48] it is still used in sequence aligning or shape matching, recently in [49,50,51]. DTW introduces elasticity into the comparison of two temporal signals, effectively mitigating the impact of small differences and misalignments while emphasizing significant changes. This metric enhances the robustness of the algorithm in real-world scenarios, mitigating the impact of small variations in shape or segmentation errors. For example, the small misalignment illustrated in Figure 6 yields a distance ten times smaller when using DTW, with a small elasticity parameter of three steps.

Employing this comparison function yields a 3D matrix of distances to the model for each view. These matrices represent the distance to the model across different object rotations. This methodology enables not only the determination of distance from an object to the model but also allows estimation of the most similar rotation.

### 3.4. View-Compatible Feature Fusion (VCFF)

To effectively fuse the outcomes from all available views, it is imperative to combine the acquired 3D matrices of distances. A direct approach might involve obtaining the minimum value from each matrix and then merging these values by computing the average, minimum, maximum, or median. However, this simplistic method would merely yield the average of the votes, the most similar camera, or the most different camera. These approaches fail to optimally merge 3D information, potentially combining rotations that are physically impossible due to camera positions.

To achieve a more robust fusion, we propose integrating pose information. Alongside each view, relative positions to other cameras are introduced. This additional information enables straightforward rotation of the 3D matrix of distances from one camera point of view to align with another camera’s perspective. Consequently, each matrix value can be seamlessly added to its counterpart in another camera’s matrix, facilitating the combination of only view-compatible features. In essence, we are combining values that are 3D feasible, ensuring that the fusion process considers the object’s shape in three dimensions.

Figure 3 presents an example of VCFF using a three-camera configuration and pseudocode is provided in Algorithm A3 in Appendix A. To make it easier to understand, only two rotation dimensions are shown. Each camera is positioned at the end of an orthogonal axis relative to an object—a rubber duck. The feature comparison results for each camera are illustrated as a matrix of images, showing the model at 90° intervals along two axes, assuming the comparison method is rotation-invariant with respect to the camera’s axis. If the feature comparison function successfully matches the camera’s point of view, a green checkmark appears in the top-right corner; otherwise, a red cross indicates no match. The central columns of the figure show how the result matrices for Cameras 2 and 3 are rotated to align with Camera 1’s point of view using relative pose information—specifically, a 90° rotation along the Z and Y axes, respectively.

Finally, the rightmost section of the figure displays the fusion results as a matrix of image trios, where each trio consists of one image from each camera at compatible positions. For simplicity, only two rotation dimensions are shown in the fusion result; however, in a complete fusion process, all three rotation dimensions would be considered, as the object is fully characterized in 3D space. Only one position achieves a full match across all three cameras for the feature comparison function.

More practically, the relative positions between cameras can be represented using transformation matrices, each mapping from each camera’s coordinates to those of a destination camera, typically the one used for training. For example, rotating the coordinate system of Camera X to align with the destination camera can be achieved by multiplying each position in the result matrix by Camera’s X transformation matrix. The result is a new position in the destination coordinate system. Repeating this procedure, destination matrix cells are filled, combining the result of all the cameras, for instance adding the result of all the compatible positions, after which the minimum can be selected as the most feasible.

As illustrated in Figure 3, consider the position (90, 90) in Camera 2. The object in this position is rotated 90° around both the Y and Z axes. By applying a transformation matrix that rotates −90° around the Z-axis, which is the relative rotation between cameras 1 and 2, this position maps to (90, 0) in the destination coordinate system, which corresponds to a 90° rotation around the Y-axis. Therefore, the final transformed position of (90, 90) is (90, 0) in the destination matrix (shown on the right side of the figure). Similarly, position (90, 0) from Camera 1 and position (180, 0) from Camera 3—which is rotated -90° around the Y-axis—are also mapped to the same cell in the destination matrix.

Cameras’ relative positions are usually obtained through a process called calibration. Calibration procedures in industrial applications are well-established and typically achieve sub-pixel accuracy such as in Perez et al. [45]. Rotations are sampled in relatively large steps—every 4° in our experimental setup—resulting in changes that correspond to several pixels. Therefore, small calibration errors on the order of a pixel are assumed to have no significant impact on the results and are considered out of the scope of this work.

In the case of using a reduced 2D matrix of distances due to rotation invariance in the camera axis, it has to be extended for 3D fusion. It can be done trivially by just copying the same distance in the third dimension, camera rotation. In any case the result of the fusion will be a 3D matrix from a common point of view, usually that of the first camera.

This approach obtains precise results when the object is centered in the image. However, if the object is not centered, an additional correction is necessary to offset the translation. In such cases, estimating the centroid becomes necessary. This can be achieved by computing the center of mass of the silhouette, determining the closest point of intersection of all centers of mass from multiple views, or ideally, deriving the centroid from a 3D reconstruction generated from multiple views. The matrix of distances must then be additionally rotated by the angle difference between the center of the image and the estimated centroid, but in the opposite direction, ensuring the matrix of distance is computed as if the object appears centered in the image.

### 3.5. Classification Using Model-Based Normalization

The output of the VCFF includes both the distance and the rotation of the object, enabling direct comparison of this distance across all potential references or models to classify the object as the closest match. However, this approach has three primary drawbacks: first, certain objects may yield higher or lower distances due to their size or geometric complexity; second, the distance units may lack sufficient interpretability to establish an effective threshold for discarding unknown objects; and third, some object rotations can be difficult to identify, introducing noise and resulting in higher distances.

To simplify the thresholding phase, we propose a normalization method that utilizes training statistics. Specifically, we estimate the distribution of distances under a worst-case scenario. The worst case scenario is defined as the matrix of positions in-between training points, as captured objects in tests will rarely be in training points and these are the cases with higher distance. So DTW distances are computed for this matrix: for each in-between training point an image is rendered, features extracted, and the DTW distance to its four neighboring trained points is measured, keeping the maximum of it. This process results in a matrix of DTW distances that measures which is the highest possible distance for each training point. Finally, these simulated distances are fused through VCFF under a specific camera configuration and then fitted to a Gaussian distribution. This process allows the derivation of the mean and standard deviation of worst-case scenarios for a specific model, providing a robust basis for normalization and thresholding.

The derived mean and standard deviation can then be used to transform DTW distances into more meaningful metrics, such as sigmas, *z*-scores or probabilities of belonging to a particular class. Moreover, this normalization process adjusts the DTW distances, which can vary significantly across models based on object size or geometric complexity.(1)$$z=\frac{x-\mu }{\sigma }$$

Consequently, the proposed output for comparing a set of images to a model is the *z*-score Equation 1 (or standard score), which expresses the distance in terms of standard deviations (sigmas) from the mean. In Equation (Equation 1) *z* is the *z* score *x* is each DTW distance to be scaled and μ and σ are the worst case mean and deviation respectively. Negative sigmas are assigned to distances below the mean. This approach effectively resolves the first two drawbacks by normalizing distances and assigning them a meaningful, interpretable and unified unit.

Additionally, inspected objects often pose challenges related to the employed metric—in this case, DTW with centroid distance. Thin, planar objects, for example, can be particularly problematic when positioned perpendicular to the camera, as small rotational variations may lead to significant changes to their silhouettes. Such rotations often produce elevated distances in almost every situation. To address this issue and improve classification performance, regions corresponding to these problematic rotations can be excluded during feature fusion. This allows the classification decision to prioritize data from cameras offering superior views.

To achieve this, the proposed method ignores the X% highest distances in the 3D matrix of simulated worst-case scenarios, ensuring that cameras with better views determine the global distance in these positions. However, with a limited number of cameras, excluding these problematic rotations risks omitting the real rotation, potentially leading to incorrect classifications. For this reason, this technique is only applied when a minimum number of cameras are available, as discussed in the results section.

The exact percent of ignored distance is highly dependant on the 3D object shape. Different statistical approaches can be used to compute the best point such as fitting to a distribution, for instance, Gaussian, and filtering points beyond three sigma. However, in this case it has been empirically set to 5% as it showed a good balance between reducing false negatives and not increasing false positives.

## 4. Results

To validate and compare the proposed solution with state of the art approaches, comprehensive experiments have been conducted. Three different algorithms have been compared: the proposed solution, RotationNet [32], and MVCNN [19]. These solutions have been selected as representative of the best-performing simultaneous and sequential methods found in the literature. Although, MVC is an active research topic and new methods are constantly proposed, MVCNN and RotationNet were chosen as solid representatives of the main strategies where most of the research line is fundamented. Furthermore, due to the specific needs of industrial inspection it was considered that well-stablished methods that effectively handle information are the best possible choice.

Whenever possible, the original code from the authors has been used to avoid implementation errors. Since only RotationNet requires a specific camera configuration, the dodecahedron configuration proposed in it has been used, as the authors state it is the best-performing alternative.

The dataset used is presented in Section 3.1. During the training stage only 11 of the 15 objects were used. All 15 objects were inspected during testing, expecting an unknown result for the objects not present in the training stage. Training images were rendered from the 3D models of the objects. Each algorithm rendered as many images as necessary for proper training: VCFF used only 4050 images per object, while RotationNet and MVCNN used 16,000 and 20,000 images per object, respectively.

The testing set consisted of 100 captures, each with 20 images, arranged in a dodecahedron configuration, resulting in a total of 2000 images per object. Each capture has been tested with 1 to 20 cameras to analyze how the metrics evolve with respect to the number of cameras used. For each camera configuration, the most perpendicular combination has been selected to maximize the amount of available information.

Two different metrics have been used to compare the performance of the three algorithms. To measure the overall performance of each algorithm, the area under the precision–recall curve (average precision) is used as it provides a comprehensive evaluation across multiple classes and handles class imbalance effectively. However, since this is an open-set multi-class problem, direct application of this metric is not feasible. It is necessary to convert each prediction output, which is a probability or distance to each trained class, into N predictions, one per trained class, represented as tuples correct-class (0, 1) and probability or distance.

Examples of this conversion can be seen in Table 2 and Table 3. These predictions can then be used to calculate the average precision for all classes, including unknown class. Table 2 represents the result of the complete VCFF for each model and three objects, units are *z*-scores. These results are transformed to Table 3 in the form of tuples correct-class (0, 1) and distance.

Figure 7 shows the average precision on the dataset without unknowns for different number of cameras. As mentioned in Section 3.5, VCFF has two variants: one that ignores 5% of the worst rotations, labeled “VCFF_ignore”, and one that considers all positions, labeled “VCFF_complete”. As expected, the complete version performs better with 1 and 2 cameras, where bad training positions are excluded, but there are not enough cameras to provide adequate information on these positions, resulting in incomplete exploration. For this reason, the following results, labeled as “VCFF”, will use the complete approach for one and two cameras and the “VCFF_ignore” approach (which ignores problematic rotations) for the rest.

As seen in the figure, the MVCNN alternative performs notably worse than the other algorithms. This is expected, as MVCNN does not technically make assumptions about 3D structure, relying solely on averaging multiple views. As a result, it struggles to distinguish mirrored configurations. Furthermore, MVCNN performs poorly in configurations with few cameras, as it was trained with a 20-camera configuration and would require retraining with the actual number of cameras for optimal performance, which is usually a problem with the long training times. In contrast, the other approaches allow the use of different camera configurations during training and testing.

RotationNet solves this problem effectively, achieving 1.0 AP with just three cameras, while VCFF performs slightly worse. This indicates that the features extracted by RotationNet, based on CNN, are more effective than the ones used in VCFF. This difference is particularly noticeable when using only a single camera.

However, the main goal of this study is the industrial inspection scenario, where it is crucial to distinguish unknown objects. Figure 8 shows the average precision results in an open-set scenario, where four unknown objects must not be misclassified. In this context, RotationNet and MVCNN are unable to achieve 1.0 precision while VCFF with eight cameras successfully accomplishes this.

As expected, MVCNN performs poorly, particularly due to its reduced efficiency when using fewer cameras than during training and its lack of coherence between views, making it impossible to achieve good results compared to the other methods. RotationNet performs better with just one camera, likely because the extracted features are more representative, but feature fusion in VCFF is clearly superior, with VCFF outperforming the others from two cameras onward.

It is also noteworthy that RotationNet’s performance appears to stabilize around 11 cameras, meaning that adding more cameras does not lead to further improvements. This is likely because the features extracted by RotationNet are better suited for distinguishing known objects rather than differentiating between known and unknown objects, a challenge that arises from the dataset bias inherent in open-set problems. VCFF does not exhibit this issue, as it is compared with the 3D model through VCFF, introduced in this work, incorporating pose information. This allows it to reject anything that deviates from the model, ensuring greater robustness and precision.

The second metric used is the area under the ROC curve (AUC) up to a maximum false positive rate of 3σ, or 0.25%. This metric is particularly important in industrial inspection, where misclassifying an object (false positive rate) represents a critical error, making the remainder of the ROC curve less relevant. In other words, it is preferable to classify an object as “unknown” rather than incorrectly identifying it as another object. Once this criterion is satisfied, the optimal algorithm is the one that achieves the highest true positive rate, correctly identifying more objects.

Figure 9 presents the results for this metric. The trends are similar to those observed in the previous metric, but more pronounced. VCFF clearly outperforms both RotationNet and MVCNN, achieving near 1.0 AUC with seven cameras, demonstrating its strong performance in the most critical metric for industrial inspection.

Finally, since the proposed method also provides pose estimation as a byproduct of the classification, the error in this estimation has also been measured. To evaluate the error, the simulated capture rotation has been compared with the estimated rotation, and the error in degrees has been calculated. It is important to note that training images are sampled at 4° intervals, and the system returns one of these predefined rotations, so the minimum mean error is 2°.

Figure 10 shows the rotation error as a function of the number of cameras used. The error with one camera has been omitted because only contour information is used, and the metric is rotation-invariant along the *z*-axis of the camera, leading to almost random guesses. However, as more cameras are introduced, the error decreases rapidly, approaching the minimum error of 2°.  

Regarding computational complexity and scalability, the proposed algorithm has both advantages and disadvantages compared to state-of-the-art alternatives. VCFF trains significantly faster, requiring only two minutes per reference on a standard PC, whereas MVCNN and RotationNet take hours to train on specialized hardware. Moreover, adding new references to the training set in the proposed algorithm simply involves adding the new 3D object trained to the set of models, whereas deep learning approaches require a complete retraining process.

However, during testing, the proposed algorithm takes less than one second per reference, while state-of-the-art methods have negligible inference times—though they rely on specialized hardware, and are independent of the number of references. These processing times can be optimized, as the proposed algorithm is highly parallelizable and can be adapted for GPU acceleration, significantly reducing runtime. Additionally, a pre-classification step can be incorporated to quickly discard clearly different objects based on size, color, or other easily computed characteristics.

## 5. Conclusions and Future Work

In this work, a multi-view classification algorithm for industrial inspection based on view-compatible feature fusion (VCFF) has been presented. In this context, VCFF outperforms state of the art methods due to its enhanced discrimination capabilities incorporating pose information. By merging information from different views, it is able to reject similar objects, even those not seen during training: open set. These advantages are particularly evident in the most important metrics for industrial inspection, such as true positive rate with almost zero false positives. This means that the algorithm correctly classifies as many objects as possible while maintaining a low rejection rate and avoiding misclassification.

The assumption that pose estimation naturally arises as a byproduct of classification is validated by the results, which show minimal pose estimation error when combining information from multiple cameras. Therefore, we conclude that including pose information during training and relative camera positions during testing is of utmost importance to achieve higher classification precision, as both tasks are closely related. Without this, multi-view classification strategies would need to infer pose information from images, which can be readily available and thus simplify the problem.

Regarding computational complexity and scalability, the proposed algorithm offers advantages in training, as it is faster to train and allows new references to be seamlessly added. Although its inference time is significantly slower compared to state-of-the-art methods, it remains fast enough for deployment in most production lines. Moreover, its performance can be further optimized through GPU acceleration if needed, or a pre-classification step can be incorporated. These optimizations are left for future work.

A key limitation of the proposed approach is the use of hand-crafted Fourier descriptors for feature extraction. These descriptors have limited generalization and discrimination capabilities compared to state of the art CNN-based features, and they rely solely on external contours, ignoring texture, internal holes, or other potentially discriminative information. However, these features do not suffer from dataset bias or other issues associated with learned features, making them more suitable for open-set problems like industrial inspection. Future work will focus on leveraging more advanced features with better discriminative power, enabling the method to handle a broader range of objects as well as real world scenarios.

## Figures and Tables

**Figure 1 sensors-25-04127-f001:**
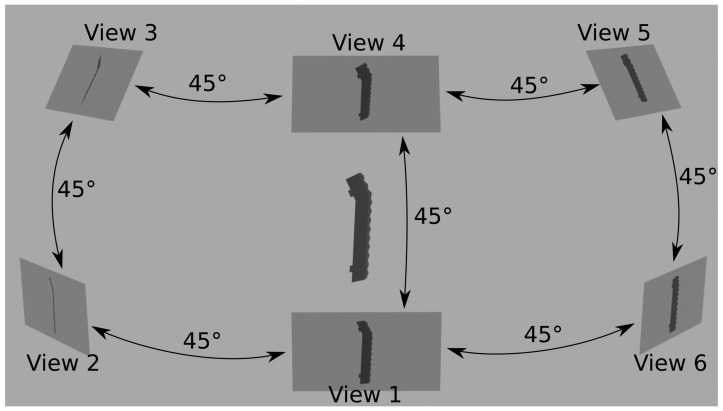
3D object shown from multiple views in order to identify it.

**Figure 2 sensors-25-04127-f002:**
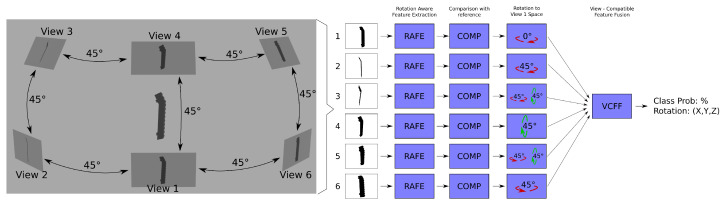
Schema of View-Compatible Feature Fusion algorithm using a 6-view camera example.

**Figure 3 sensors-25-04127-f003:**
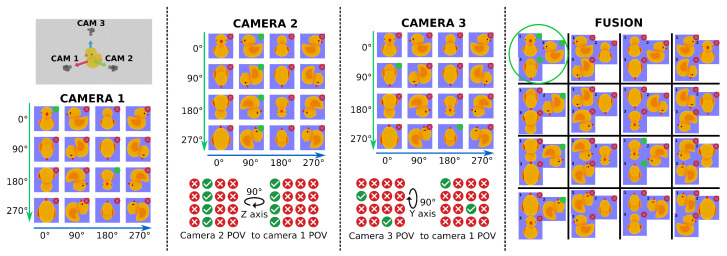
Example of View-Compatible Feature Fusion (VCFF) with three cameras and two rotation dimensions. From left to right: (1) 3D camera and object placement, (2) feature comparison result for Camera 1, (3) feature comparison result for Camera 2 with rotation to Camera 1’s point of view, (4) feature comparison result for Camera 3 with rotation to Camera 1’s point of view, and (5) fusion results.

**Figure 4 sensors-25-04127-f004:**
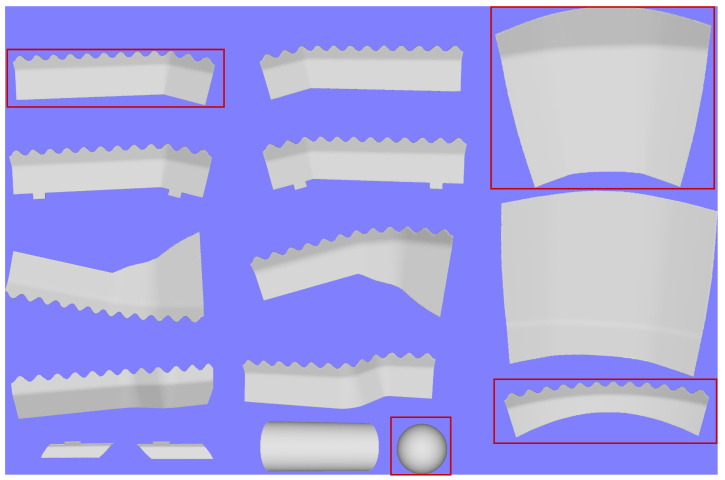
Objects to be classified in the dataset. Red outlined objects are NOT included in the training set, so they must be classified as unknown.

**Figure 5 sensors-25-04127-f005:**
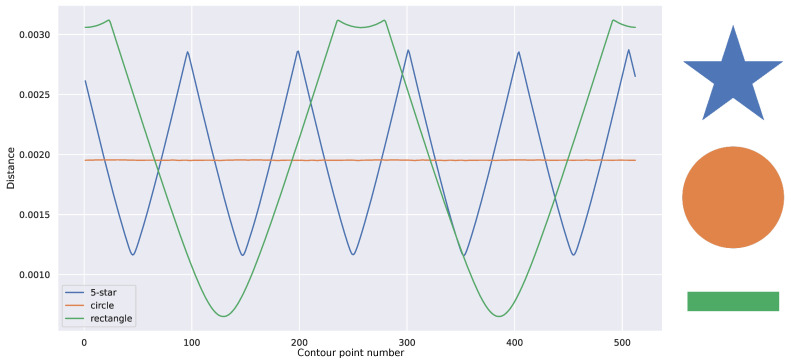
Graph of 1D central distance Fourier descriptor extracted from image shapes (**left**), alongside corresponding source images (**right**) used for feature extraction: 5-pointed star, circle, and rectangle.

**Figure 6 sensors-25-04127-f006:**
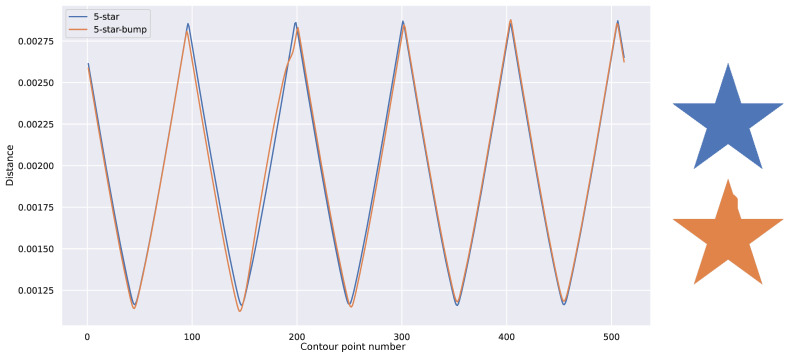
Graph showing misalignment in signals due to small changes in the contour (**left**), alongside corresponding source images (**right**) used for feature extraction.

**Figure 7 sensors-25-04127-f007:**
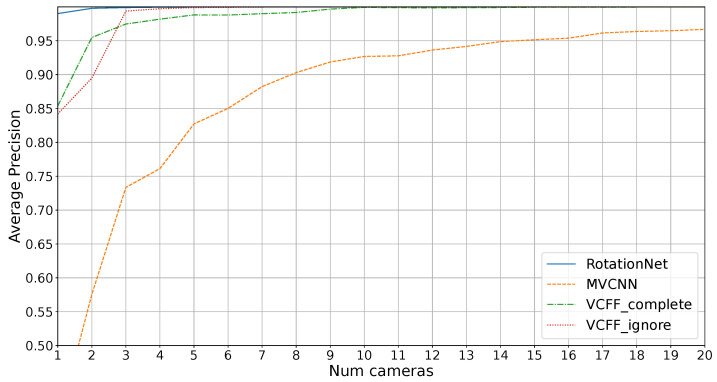
Results for average precision on diferent number of cameras on closed set, no unknowns.

**Figure 8 sensors-25-04127-f008:**
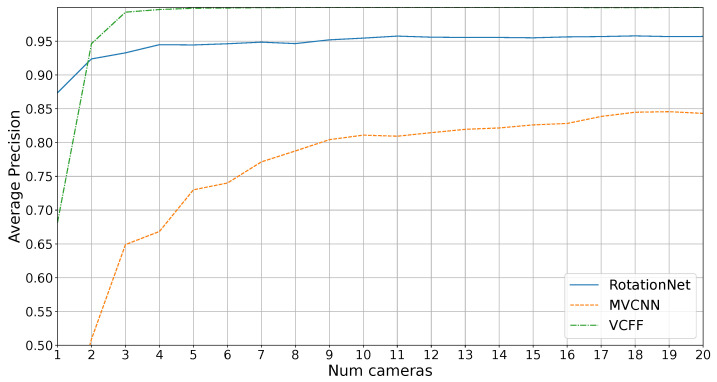
Results for average precision on diferent number of cameras on open set.

**Figure 9 sensors-25-04127-f009:**
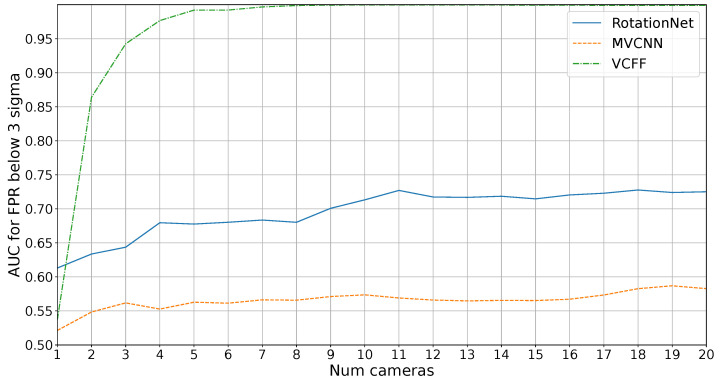
Results for area under the ROC curve on different numbers of cameras on open set.

**Figure 10 sensors-25-04127-f010:**
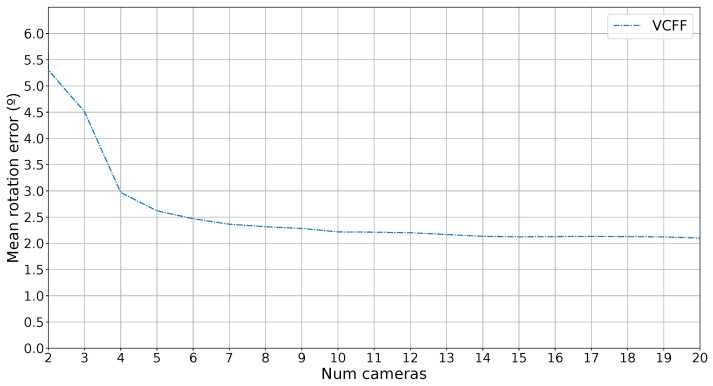
Rotation error for VCFF while classifying.

**Table 1 sensors-25-04127-t001:** Multiview 3D models classification dataset information.

Name	Type	Description	# Instances	Total Instances	Train/Test
3D model	.ply	Triangle mesh of the 3D geometry of the object	11	11	TRAIN
Calibration	.ini	Camera configuration in test	1	1	TEST
Object	Multiple	Group of captures of an object to classify	15	15	TEST
Capture	Multiple	Group of images and rotation of an object instance to classify	100 per object	1500	TEST
Images	.png	Simulated images of an object in a capture	20 per capture	30,000	TEST
Rotation	binary	Rotation of the 3D model wrt world in a capture	1 per capture	1500	TEST

**Table 2 sensors-25-04127-t002:** Example of distances (*z*-score) to 3 models for 3 diferent objects corresponding to model1, model2, and an unknown that needs to be converted to obtain average precision, see Table 3.

	Model 1	Model 2	Model 3
Object1	−2.1	4.3	12.1
Object2	5.1	−2.3	12.6
Uknown	15.1	12.3	12.8

**Table 3 sensors-25-04127-t003:** Predictions from Table 2 converted so average precision can be easily computed.

	Class	Distance (*z*-Score)
Obj1 vs. Model 1	1	−2.1
Obj1 vs. Model 2	0	4.3
Obj1 vs. Model 3	0	12.1
Obj2 vs. Model 1	0	5.1
Obj2 vs. Model 2	1	−2.3
Obj2 vs. Model 3	0	12.6
Ukn vs. Model 1	0	15.1
Ukn vs. Model 2	0	12.3
Ukn vs. Model 3	0	12.8

## Data Availability

The dataset created for this work is publicly available at Zenodo (https://zenodo.org/records/15058296, accessed on 29 June 2025).

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
