# Peer review of "Object-Specific Multiview Classification Through View-Compatible Feature Fusion"

_sensors, 2025, doi:10.3390/s25134127_

Round 1
Reviewer 1 Report
Comments and Suggestions for Authors
This manuscript presents a method to do multi-view classification MVC. In this method, a View-Compatible Feature Fusion VCFF is proposed. But still there are some suggestions and comments, as following:
1、Since in the manuscript, the authors mentioned a lot about industrial objects, it is suggested that target industrial objects are given and discuss how the method addresses their distinct geometric differences. Meanwhile, it is better that the term "open-set" is given a definition.
2、Again, the experimental dataset seems relatively small, which may affect the significance and generalizability of the method. And it seems that the models are based on virtual objects, which may not fully capture the complexities of real-world scenarios. In real industrial application, real parts are not always the same with CAD models. It is better if at least some discussion or experiment on real part are given.
3、In this manuscript, the method for fusing pose/angle information into the framework could be seen as an important innovation. But it is not clearly explained. it is suggested that clarifying this would help improvng the paper's innovation
4、It is suggeste that the authors should elaborate on how the method handles subtle geometric variations (e.g., sub-millimeter differences or mirrored parts) which is very common in industrial manufacturing scenarios especially for coarse machining.
5、 (Sec. 3.2): The presentation of viewpoint/pose parameters is inconsistent or incomplete. A standardized format (e.g., Euler angles, quaternions) is recommended. And, the authors claim to integrate pose information but it could be improved to describe how it is computed.
6、(Sec. 3.3) "As previously mentioned, owing to rotation invariance, this 3D matrix can be trivially reduced to 2D" We doubt this, he description of 3D rotation invariance is problematic. While 2D shapes are rotation-invariant, 3D pose estimation is inherently more complex. We suggest a better descrip
7、In this manuscript, the proposed method heavily depends on Fourier descriptors. Since Fourier descriptors are centroid-based and primarily effective for 2D shape analysis. Especially for shapes even 2D shapes with internal holes or features. So we suggest authors to better discuss about this.
8、The paper relies on Dynamic Time Warping (DTW) but cites outdated references (1959). Maybe some new references could be added.
9、In the section 3.5, The rationale for ignoring the top X% distances in simulated worst-case scenarios is unclear. How is X determined? Is it empirical or theoretical?
10、The relationship between Tables 1–2 and preceding figures is not explained. A discussion linking these results is needed.
11、 The calibration process and its error margins are not discussed, Would the authors talk about its influence of recognition accuracy This is critical for industrial applications.
Reviewer 2 Report
Comments and Suggestions for Authors
- The manuscript contains information about the state of the art, but there is no information about the broad discussion of multi view classification (MVC) and pose estimation technologies in the broad scope of the applications. The introduction is started from the description of the specific problem. And it is described in general, that examples of the specific tasks, where the MVC is required, are not provided. These two problems should be addressed. For the addressing the first problem I can recommend to describe the applications of the MVC and pose estimation technology in such areas as a sport [https://doi.org/10.1109/ICDSC.2009.5289407, https://doi.org/10.1016/j.entcom.2024.100761], surveillance [https://doi.org/10.1007/s11042-018-7108-9], entertainment [https://doi.org/10.1109/TVCG.2005.105], medicine [https://doi.org/10.3390/fi13070173], forensic sciences [https://doi.org/10.1364/JOSAA.478498] and human-computer interaction [https://doi.org/10.1109/THMS.2015.2467212]. For the second problem I recommend providing the specific examples within the narrow direction, on which you decided the focus.
- In the discussion of the state of the arts you focused on deep neural networks, but ignore the historical context and the methods which were developed and applied before the wide distributions of neural networks. I believe that inclusion of such historical is necessary for the high ranked journal, such as Sensors, which have a broad auditory of readers and could be interested in understandings of more general research landscape.
- In the discussion of the state of the art you overlooked the recent works on the pose estimations. E.g. on the Sensors journal recently several related papers have been published:
- https://doi.org/10.3390/s24113427
- https://doi.org/10.3390/s25051513
- Finally in the discussion of the state of the art you did not mention competing emerging computational approaches, which still maybe not mature enough, but have some advantages over the classical machine vision based on the camera with classic optical systems. Namely, one of them is the ghost imaging. Currently, it allows recognizing the objects at an earlier stage of data acquisition, without forming a whole image [http://dx.doi.org/10.2139/ssrn.5127591]. It also has opportunities for 3D imaging [https://doi.org/10.1364/AO.492208]. So, its conjugation with neural network analysis should be of interest.
- The structure of the manuscript can be further optimized: E.g. in lines 144-145 you mentioned that your technique is aimed on the industrial applications. But this is better to discuss at the end of the introduction.
- Please describe how your technique is working with the object which has a serious occlusion during the capturing. I recommend formulating any qualitative limitations.
- Pleasedescribe how your technique works with the objects, which parts are seriously occluded during the shooting. I recommend formulating any qualitative limitations.
- Figure four is very unclear. What is a horizontal axis? How does the left part of the figure is connected to the right part?
- Please describe more explicitly whether it is implied since your objects only could be rotated or also some size scaling can be taken into account in your technique.
- Table 1, 2 you mentioned distances, but what are the units: mm, m, etc.? Please specify.
- 7 contains too small text, and it ineffective use space. Use logarithmic scale or just remove empty space in the lower part of the graph.
- Another minor comments:
- Line 171. Please remove redundant space before “ :”.
- Line 177. Hyper reference after “Zenodo” have number 1, so, it is seems that it mention the reference number 1, but is it a link on a footnote at the bottom of the page. I recommend making them more distinguishable or moving footnote to the reference list.
Reviewer 3 Report
Comments and Suggestions for Authors
Title: Object-Specific Multiview Classification Through View-Compatible Feature Fusion
This work introduces a View-Compatible Feature Fusion (VCFF) method that utilizes images from predetermined positions as an accurate solution for multi-view classification of specific objects. The work is good, and the authors have presented their work in detail; however, this manuscript currently has major and minor corrections (describe below), that should be considered carefully.
Remarks to Author(s): Please see the full comments.
1-The abstract requires a brief overview of the proposed methodology, stating the rate of improvement in the achieved results. Where, the abstract did not mention the rate of improvement achieved compared to other works in the same field. Besides, what metrics were used for evaluation?
2- The contributions of the proposed work are not clearly defined. Please list and clearly highlight the most significant research contributions.
3- The introduction section provides a brief foundation on the topic of this research; therefore, some other recent information should be added to this section with much in-depth discussion.
4- It is recommended to combine sections 1 and 2 and transfer the work related to the proposed work to a new related work section with adding more recent works, while avoiding mentioning the proposed work in it.
5- This manuscript contains many grammatical and linguistic errors, and these errors lead to a decrease in the reader's understanding, so the manuscript should be reviewed carefully.
6- It is stated that "the proposed multi-view classification approach is primarily applied to industrial inspection..." Does this mean that the proposed work cannot be used in other applications? Please justify this point with more discussion about other cases.
7- Any information, figure, equation, or dataset taken from a previous source must be cited as a reliable source, unless it relates to the authors. Please, check this issue.
8- Since a new dataset has been created taking as example industrial parts. Therefore, it is required to provide more details about it in Tabular form.
9- The adoption of Fourier Descriptors based on centroid distance needs further justification where other shape descriptors can be used that may provide higher performance.
10- Why does the proposed normalization method involve measuring the DTW distance to simulate the worst-case scenario? Please explain more clearly based on the worst-case scenario.
11- The proposed work has been compared to two other works in 2015 and 2019. It is recommended to compare it with other recent works to show its robustness. However, it is mentioned that they are best-performing simultaneous and sequential methods found in the literature, but the proposed work can be compared to other recent work in the multi-view object classification field.
12- For Figure 7, RotationNet solves the problem effectively, achieving 1.0 AP with just 3 cameras while VCFF performs slightly worse since it requires about 10 cameras to achieve good results, which is a big difference. Can authors use better feature extraction algorithms to improve performance?
13- It is recommended to shorten the conclusion to a single paragraph. In any scientific research, the conclusion should include the topics and data of the proposed work, summarize its main points, discuss its significance, and discuss future work. Please review all of these points to write the conclusion section.
Comments on the Quality of English LanguageThe English could be improved to more clearly express the research.
Reviewer 4 Report
Comments and Suggestions for Authors
The authors introduced a View-Compatible Feature Fusion (VCFF) method that utilizes images from predetermined positions as a solution for multiview classification of specific objects. The experimental show that the proposed VCFF method outperforms several MVC algorithms. Before publication, some comments are provided to improve the quality of this manuscript.
- How to combine features from multiviews efficiently using VCFF? More explanations are necessary.
- There are many multiview classification available in literatures. The introduction part is not informative. For benefits of readers, the authors need to introduce and discuss related references, e.g., “Multiview generative adversarial network and its application in pearl classification. IEEE Transactions on Industrial Electronics, 2019, Vol. 66(10): 8244-8252.”. What is the main difference between VCFF and existing multiview classification models?
- Suggest the authors add more cases for further investigation.
Round 2
Reviewer 1 Report
Comments and Suggestions for Authors
Acceptable after some improvements.
Several comments:
1、Innovation of the method could be given more clearly to improve the scientific soundness.
2、Effective Mean Square Differences: A Matching Algorithm for Highly Similar Sheet Metal Parts https://www.mdpi.com/2443386 could be added as reference. Sensors 2023, 23(16), 7300; https://doi.org/10.3390/s23167300. Which are the real images of parts.
3、The illustrated examples is better to be consistent from the beginning to the end.
4、For the method, a diagram showing the overall schema could be given, which would attract the readers and give them more vivid understanding of the method. Especially for the part of VCFF.
5、Dataset and code could be given for better validation.
6、More experiment comparisons with the State of the art methods could be given if possible.
Reviewer 2 Report
Comments and Suggestions for Authors
Thanks for the careful revision of the work. Now its quality became better, and in general the manuscript already remind the work, which can be published. But a few remaining concerns still take place:
- In the revised introduction you discussed the application areas. Then you wrote: "In Xuan et al", This is an unfortunate wording, since it seems, that Xuan is also application area. I recommend to rewrite it as "In the work of Xuan et al ...".
- Please provide an equation for z-score, how you estimate it and the reference.
- The text on Figs. 7-10 is still small. Please increase the font size.
Reviewer 3 Report
Comments and Suggestions for Authors
Most of the comments have been addressed correctly. However, there are still some minor points that should be processed which are as follows:
1- The table titles should be above them.
2- It is recommended that the conclusion section be shorter and written in one paragraph.
Reviewer 4 Report
Comments and Suggestions for Authors
Most comments have been responded.
